# Systemic Immune Inflammation Index (SII), System Inflammation Response Index (SIRI) and Risk of All-Cause Mortality and Cardiovascular Mortality: A 20-Year Follow-Up Cohort Study of 42,875 US Adults

**DOI:** 10.3390/jcm12031128

**Published:** 2023-01-31

**Authors:** Yiyuan Xia, Chunlei Xia, Lida Wu, Zheng Li, Hui Li, Junxia Zhang

**Affiliations:** 1Department of Cardiology, Nanjing First Hospital, Nanjing Medical University, Nanjing 210006, China; 2Department of Intensive Medicine, The Affiliated Jiangning Hospital of Nanjing Medical University, Nanjing 211166, China

**Keywords:** National Health and Nutrition Examination Survey (NHANES), inflammation, all-cause mortality, cardiovascular mortality, systemic immune inflammation index, system inflammation response index

## Abstract

Background and aim: Chronic low-grade inflammation is associated with various health outcomes, including cardiovascular diseases (CVDs) and cancers. Systemic immune inflammation index (SII) and system inflammation response index (SIRI) have lately been explored as novel prognostic markers for all-cause mortality and cardiovascular mortality. However, studies on prediction value in nationwide representative population are scarce, which limit their generalization. To bridge the knowledge gap, this study aims to prospectively assess the association of SII, SIRI with all-cause mortality and cardiovascular mortality in the National Health and Nutrition Examination Survey (NHANES). Methods: From 1999 to 2018, 42,875 adults who were free of pregnancy, CVDs (stroke, acute coronary syndrome), cancers, and had follow-up records and participated in the NHANES were included in this study. SII and SIRI were quantified by calculating the composite inflammation indicators from the blood routine. To explore the characteristics of the population in different SII or SIRI levels, we divided them according to the quartile of SII or SIRI. The associations between SII, SIRI, and all-cause mortality and cardiovascular mortality events were examined using a Cox regression model. To investigate whether there was a reliable relationship between these two indices and mortalities, we performed subgroup analysis based on sex and age. Results: A total of 42,875 eligible individuals were enrolled, with a mean age of 44 ± 18 years old. During the follow-up period of up to 20 years, 4250 deaths occurred, including 998 deaths from CVDs. Cox proportional hazards modeling showed that adults with SII levels of >655.56 had higher all-cause mortality (hazard ratio [HR], 1.29; 95% confidence interval [CI], 1.18–1.41) and cardiovascular mortality (HR, 1.33; 95% CI, 1.11–1.59) than those with SII levels of <335.36. Adults with SIRI levels of >1.43 had higher risk of all-cause (HR, 1.39; 95% CI, 1.26–1.52) and cardiovascular death (HR, 1.39; 95% CI, 1.14–1.68) than those with SIRI levels of <0.68. In general population older than 60 years, the elevation of SII or SIRI was associated with the risk of all-cause death. Conclusion: Two novel inflammatory composite indices, SII and SIRI, were closely associated with cardiovascular death and all-cause death, and more attention should be paid to systemic inflammation to provide better preventive strategies.

## 1. Introduction

Cardiovascular diseases (CVDs), especially coronary heart disease, cerebrovascular disease, and peripheral arterial vascular disease, are complex chronic diseases, with plenty of risk factors including dyslipidemia, hypertension, insulin resistance, hypercoagulability, and inflammatory responses [1]. Even though great advances have been achieved in the prevention and treatment of CVDs, they remain the leading causes of death and disabilities worldwide [2]. A better understanding of the clinical predictors for the development and worsening of CVDs is an unmet need to be able to deliver new insight on earlier prevention and more innovative management.

To reduce the haunting threat of CVDs on public health, researchers have increasingly focused on the role of low-grade inflammatory injuries in the disease process [3]. There is evidence that patients with systemic inflammatory diseases are at a much higher risk of developing CVDs than the general population [4], indicating that inflammatory variables could be applied as predictive surrogates for CVDs. In acute coronary syndrome, cytokines such as interleukin, as well as white blood cell subsets, cause endothelial dysfunction and have been utilized for risk stratification [5,6]. It is reported that proinflammatory cell subsets secrete more interferons and chemokines, rendering the balance in lymphocyte subsets disrupted [7]. Atherosclerotic plaques are susceptible to rupture and form thromboses due to intensified inflammation. Monitoring peripheral blood is a simple but meaningful tool considering elevated blood inflammatory indicators are closely related to disease progression [8,9]. In clinic, white blood cell counts and subset counts from blood routine tests [10,11], as well as acute phase proteins [12], are used as surrogates for systemic inflammation. Furthermore, immunological inflammatory cells have been found to be predictive factors related to the occurrence and prognosis of CVDs. The neutrophil-to-lymphocyte ratio (NLR), platelet-to-lymphocyte ratio (PLR), and monocyte-to-lymphocyte ratio (MLR) have already been calculated from blood routine tests and extensively studied as prognostic tools. Their predictive value for all-cause mortality and cardiovascular mortality is better than white blood cell counts, and subset counts directly obtained from blood routine tests [13,14].

The systemic immune inflammation index (SII) and system inflammation response index (SIRI), two novel composite indices integrating three independent white blood cell subsets and platelets, are reminiscent of the interaction of thrombocytosis, inflammation, and immunity. Recent study has suggested that SII is related to the incidence of cerebrovascular diseases in Chinese adults [15]. In addition, the Kailuan study, a prospective cohort study based on the Kailuan community in the city of Tangshan, Hebei province, China, has revealed the associations of SII and SIRI with the risks for CVDs and all-cause mortality [16,17]. It thus represents the health status and potential exposures of local people living in the northern part of China. The NHANES database is a nation-wide representative cohort with different ethnicities, using a complex, multistage, probability sampling design. The conclusions drawn from the NHANES database can be more robustly generalized in reality. In this study, we investigate whether novel SII and SIRI indices from the NHANES database can be prognostic markers for CVDs and all-cause mortalities in the general population.

## 2. Materials and Methods

### 2.1. Study Design and Participants

This is a prospective cohort study. NHANES is intended to provide nutrition and health information about US adults and children to policymakers. It is updated regularly and free to the public. Detailed methods can be extracted from the official website (http://www.cdc.gov/nchs/nhanes.htm, accessed on 1 November 2022). Written informed consents were provided for participants in NHANES. In addition to that, the entire study was certified and approved by the Centers for Disease Control and Prevention’s Institutional Review Board. In all, 101,316 enrolled participants were surveyed in ten consecutive NHANES circles from 1999/2000 to 2017/2018. Participants were excluded from this study if they met at least one of the following conditions: (i) missing data on complete blood routine count; (ii) age < 18 years; (iii) baseline pregnant; (iv) with baseline CVDs such as stroke or acute coronary syndrome; (v) with baseline cancer; (vi) without follow-up records (Figure 1).

### 2.2. Definition of SII and SIRI and Classification into Groups

In this study, the definition of two inflammation-based parameters was previously described [18]. The values of SII and SIRI were first analyzed as continuous variables. There is no established standard for grouping SII and SIRI, according to previous reports. In an updated meta-analysis, a total of 13 studies were eligible for analysis [19]. In these studies, the determination of SII cutoff values varied and was based on Receiver Operating Characteristic (ROC) analysis, Youden index, and quartiles. In Xu’s study [15], they determined the best cutoff values for evaluating SII levels and the incidence of CVDs in middle-aged and elderly Chinese based on quartiles. In the study of Lin et al. [20], they divided SII and SIRI into four groups based on quartiles, and evaluated the association between the two indices and the adverse prognosis of atrial fibrillation-associated ischemic stroke patients. Based on previous studies, according to SII or SIRI distribution, participants were equally classified into four groups: lower SII (Q1, <335.36) or SIRI (Q1, <0.68), low middle SII (Q2, 355.36–468.83) or SIRI (Q2, 0.68–0.98), middle SII (Q3, 468.84–655.55) or SIRI (Q3, 0.99–1.42) and high SII (Q4, >655.56) or SIRI (Q4, >1.43).

### 2.3. Determination of Mortality Outcomes

To determine mortality status in the follow-up population of NHANES, in this study publicly-accessible death data were used until 31 December 2019 (https://www.cdc.gov/nchs/data-linkage/mortality-public.htm, accessed on 1 November 2022). Statistical outcomes were determined according to ICD-10, the International Statistical Classification of Diseases, 10th Revision: all-cause death, CVDs death (ICD-10: 054-068), and cancer death (ICD-10: 019-043) [21].

### 2.4. Covariates

The demographic characteristics of age, sex, race/ethnicity, education level and family poverty income ratio (PIR) were obtained from the demographic questionnaires. Among them, age (years) was used as a continuous variable. Sex was classified as female or male. The race/ethnicity categories were classified as non-Hispanic White, non-Hispanic Black, Mexican American, other Hispanic, and others. Education was divided into three levels: below high school, high school, and above high school, in line with a previous study [22]. The poverty income ratio was used to measure poverty as defined by the US Census Bureau and a PIR of less than 1 for a particular household was in that family’s range according to the NHANES website. Risk factors such as hypertension and diabetes history, smoking and drinking status were also adopted from the health questionnaires. Body mass index (BMI) was measured and categorized into: <25.0, 25.0–29.9, and >29.9 kg/m^2^ according to the definition of obesity and overweight [23]. For smoking and drinking status, participants were categorized as never smokers, former smokers, or current smokers and as non-drinkers, low-to-moderate drinkers, or heavy drinkers as suggested by NHANES. Metabolic equivalent levels were adjudicated by published literature [24], and physical activity was divided into three categories: inactive, insufficiently active, and active. Energy intake was used as a continuous variable. After at least 8 h of an overnight fast, white blood cell (WBC), platelet, neutrophil, lymphocyte, monocyte, alanine aminotransferase (ALT) and aspartate aminotransferase (AST) levels were measured using blood samples as suggested by NHANES detailed laboratory procedures. The estimated glomerular filtration rate (eGFR) was calculated from a creatinine equation [25]. Altogether, the NHANES website provided detailed procedures in collecting blood biochemical measurements.

### 2.5. Statistical Analysis

For data analysis, IBM SPSS Statistics 26 (IBM Corporation, New York, NY, USA) and R 4.1 (R Foundation for Statistical Computing, Vienna, Austria) were used. Mean and standard deviation (SD) were used for descriptive analysis if the distribution of continuous variables were normally distributed; otherwise, medians (interquartile ranges) were utilized to describe non-normal distribution. Absolute frequencies and percentages were used to report categorical variables. Overall, the independent samples *t*-test was used to compare normally distributed continuous variables. Mann-Whitney U test or Kruskal-Wallis test was used to analyze non-normally distributed data. SII and SIRI scores were initially analyzed as continuous variables. For the sake of determining the predictive value of the two novel indices (SII and SIRI) to outcomes (all-cause mortality and cardiovascular mortality), multivariable Cox proportional hazards regression was used. To exclude confounding effects, Model 1 was adjusted for age, sex, race/ethnicity, education level, and family poverty income ratio. Model 2 was adjusted as Model 1, plus drinking status, smoking status, BMI, physical activity, total energy intakes, eGFR, ALT, AST, self-reported hypertension, and self-reported diabetes. Statistical significance was set at a two-sided *p* < 0.05.

## 3. Results

### 3.1. Baseline Characteristics of Population Stratified by SII/SIRI

Upon excluding participants younger than 18 years of age, pregnant women, people with CVDs and cancers, as well as those without follow-up data, a total of 42,875 adults were enrolled in the study. The mean age of the adult participants was 44 ± 18 years old and 47.8% were male. The study cohort consisted of 8578 Mexican American (20.0%), 3727 other Hispanic (8.7%), 17,029 non-Hispanic White (39.7%), 9287 non-Hispanic Black (21.7%), and 4254 other race (9.9%). In addition, 70.0% of the participants were low-to-moderate drinkers. The proportions reporting hypertension and diabetes were 27.0% and 8.8%, respectively. At the median 9.83-year follow up, 4250 (9.91%) participants had died. A total of 998 (2.33%) participants died from CVDs. Further, the baseline characteristics of participants are summarized (Appendix A). In Table 1, according to the quartiles of SII, participants were equally classified into four groups: lower SII (Q1, <335.36), low middle SII (Q2, 355.36–468.83), middle SII (Q3, 468.84–655.55) and high SII (Q4, >655.56). Participants in the Q4 group according to SII levels were more likely to be female, non-Hispanic White, former or current smokers and heavy drinkers, more likely to be obese, lack exercise, have lower energy intake, and have lower education levels. Notably, high SII levels were significantly associated with elevated levels of WBC, neutrophil, monocyte, and platelet counts, as well as lower levels of eGFR, ALT, AST and lymphocyte count. Participants who had high SII levels were more likely to be comorbid with hypertension and diabetes.

Similarly, SIRI was divided into four groups according to quartiles: lower SIRI (Q1, <0.68), low middle SIRI (Q2, 0.68–0.98), middle SIRI (Q3, 0.99–1.42) and high SIRI (Q4, >1.43) (Table 2). Similar to the characteristics of those with high SII levels, participants who had high SIRI levels were more likely to be non-Hispanic White, former or current smokers and heavy drinkers, more likely to be obese, lack exercise, and have lower education levels. They were also significantly associated with elevated levels of WBC, neutrophil, monocyte, and platelet counts, and lower levels of eGFR and lymphocyte count. Participants with high SIRI levels, however, tended to have elevated or similar ALT and AST levels compared to those with lower SIRI levels. Participants who had high SII levels were more likely to be comorbid with hypertension and diabetes. High SIRI levels were also more often noticed in older and male participants. Additionally, we found that SII and SIRI levels increased in conjunction with each other.

### 3.2. Prediction of All-Cause Mortality and Cardiovascular Mortality with SII

Two Cox proportional hazards regression models were applied to investigate the independent prediction power of SII levels in all-cause mortality and cardiovascular mortality. As shown in Table 3, in the unadjusted model (Crude), participants in the Q4 group (the high quartile of SII level >655.56) had a 32% increased risk of cardiovascular mortality (hazard ratio [HR], 1.32; 95% confidence interval [CI], 1.11–1.57) and a 31% increased risk of all-cause mortality (HR, 1.31; 95% CI, 1.21–1.43) compared to those in the Q1 group (the lower quartile of SII level <335.36). After adjusting for age, sex, race/ethnicity, education level and family poverty income ratio (Model 1), the increased risks remained for participants in the Q1 group (HR, 1.43; 95% CI, 1.91–1.71; HR, 1.38; 95% CI, 1.26–1.50, respectively). Next, after further adjustment for drinking status, smoking status, BMI, physical activity, total energy intakes, eGFR, ALT, AST, self-reported hypertension or diabetes (Model 2), the risks for cardiovascular mortality and all-cause mortality were elevated in parallel with an increase in SII, with all *p* for trend < 0.001. However, in the Q2 and Q3 groups (Q2, SII levels 355.36–468.83; Q3, SII levels 468.84–655.55), the associations were not significant in regression models. SII level was an independent risk factor for cardiovascular (HR, 1.33; 95% CI, 1.11–1.59) and all-cause mortalities (HR, 1.29; 95% CI, 1.18–1.41) in Q4 participants. Aside from that, Kaplan-Meier curves for survival show that participants in the Q4 group had a worse prognosis during follow-up, with *p* for log-rank test < 0.001 (Figure 2A,B).

Model 1 was adjusted for age (continuous), sex (male or female), race/ethnicity (Mexican American, Other Hispanic, Non-Hispanic White, Non-Hispanic Black or Other), education level (below high school, high school, or above high school), and family poverty income ratio (<1.0, or ≥1.0).

Model 2 was adjusted as Model 1 plus drinking status (non-drinkers, low-to-moderate drinkers, or heavy drinkers), smoking status (never smokers, former smokers, or current smokers), BMI (<25.0, 25.0–29.9, or >29.9), physical activity (inactive, insufficiently active, or active), total energy intakes (in quartiles), estimated glomerular filtration rate (continuous), alanine aminotransferase (in quartiles), aspartate aminotransferase (in quartiles), self-reported hypertension (yes or no), and self-reported diabetes (yes or no).

### 3.3. Prediction of All-Cause Mortality and Cardiovascular Mortality with SIRI

Similarly, we assessed whether SIRI levels could predict all-cause mortality and cardiovascular mortality (Table 4). In the unadjusted model (Crude), participants in the Q3 and Q4 groups (Q3, SIRI 0.99–1.42; Q4, SIRI >1.43) had 35% and 126% increased risks of cardiovascular mortality (HR, 1.35; 95% CI, 1.12–1.64; HR, 2.26; 95% CI, 1.89–2.70, respectively). Indeed, 43% and 118% higher risks of all-cause mortality were discerned in the two groups (HR, 1.43; 95% CI, 1.30–1.57; HR, 2.18; 95% CI, 2.00–2.38, respectively). In the adjusted Model 1, participants in the Q4 group had increased CVDs mortality and all-cause mortality risks (HR, 1.60; 95% CI, 1.32–1.94; HR, 1.57; 95% CI, 1.43–1.72, respectively). After adjusting for the cardiovascular risk factors (Model 2), results demonstrated that SIRI was an independent risk factor for cardiovascular mortality (HR, 1.39; 95% CI, 1.14–1.68) and all-cause mortality (HR, 1.39; 95% CI, 1.26–1.52). Nevertheless, predictions for cardiovascular mortality risk by SIRI were underpowered for participants in Q2 and Q3 groups in both Models 1 and 2, but still significant in predicting all-cause mortality risk (HR, 1.22; 95% CI, 1.11–1.35; HR, 1.16; 95% CI, 1.05–1.28, respectively). Survival time was significantly reduced in the Q3 and Q4 groups during follow-up (Figure 2C,D), and shortest in the Q4 group, with *p* for log-rank test <0.001.

Model 1 was adjusted for age (continuous), sex (male or female), race/ethnicity (Mexican American, Other Hispanic, Non-Hispanic White, Non-Hispanic Black or Other), education level (below high school, high school, or above high school), and family poverty income ratio (<1.0, or ≥1.0).

Model 2 was adjusted as Model 1 plus drinking status (non-drinkers, low-to-moderate drinkers, or heavy drinkers), smoking status (never smokers, former smokers, or current smokers), BMI (<25.0, 25.0–29.9, or >29.9), physical activity (inactive, insufficiently active, or active), total energy intakes (in quartiles), estimated glomerular filtration rate (continuous), alanine aminotransferase (in quartiles), aspartate aminotransferase (in quartiles), self-reported hypertension (yes or no), and self-reported diabetes (yes or no).

### 3.4. Subgroup Analysis

Subgroup analysis was performed in accordance with stratification of the population by sex and age. As shown in Appendix A, only the highest quartile level of SII predicted CVDs mortality and all-cause mortality when compared with the lowest quartile level of SII. Similar predictive capacities for increased risk of CVDs and all-cause mortality by increased baseline SII levels were seen in both males and females, regardless of age. The analysis stratified by sex revealed that compared with the Q1 group, females in the Q4 group had a 1.38-fold (95% CI, 1.06–1.81) higher risk of cardiovascular mortality and a 1.33-fold (95% CI, 1.17–1.52) higher risk of all-cause mortality, respectively. Furthermore, males in the Q4 group had a 1.30-fold (95% CI, 1.01–1.67) higher risk of cardiovascular mortality and a 1.26-fold (95% CI, 1.12–1.42) higher risk of all-cause mortality. Moreover, compared with the Q1 group, females in the Q4 group had a 1.32-fold (95% CI, 1.01–1.72) higher risk of cardiovascular mortality and a 1.40-fold (95% CI, 1.23–1.60) higher risk of all-cause mortality, respectively. It is worth mentioning that males in the highest SIRI group had a 1.46-fold (95% CI, 1.10–1.94) higher risk of cardiovascular mortality and a 1.38-fold (95% CI, 1.21–1.58) higher risk of all-cause mortality. We did not find significant interaction of SII, SIRI and sex for either cardiovascular mortality or all-cause mortality (*p* > 0.05 for interaction; Appendix A).

The analysis stratified by age revealed that participants aged <60 years in the Q4 group versus the Q1 group were associated with a 1.31-fold (95% CI, 0.88–1.95) higher risk of cardiovascular mortality and a 1.06-fold (95% CI, 0.90–1.25) higher risk of all-cause mortality. Moreover, participants aged ≥ 60 years in the Q4 group versus the Q1 group had a 1.33-fold (95% CI, 1.08–1.64) higher risk of cardiovascular mortality and a 1.39-fold (95% CI, 1.26–1.54) higher risk of all-cause mortality. Furthermore, participants aged <60 years in the Q4 group versus the Q1 group were associated with a 1.39-fold (95% CI, 0.95–2.04) higher risk of cardiovascular mortality and a 1.12-fold (95% CI, 0.94–1.32) higher risk of all-cause mortality. Participants aged ≥ 60 years in the Q4 group versus the Q1 group had a 1.56-fold (95% CI, 1.24–1.95) higher risk of cardiovascular mortality and a 1.66-fold (95% CI, 1.48–1.86) higher risk of all-cause mortality (Appendix A). In addition, age was a significant modifier (*p* = 0.008 and 0.003 for interactions, respectively) for the association of SII and SIRI levels with all-cause mortality (Appendix A).

### 3.5. Sensitivity Analysis

We performed sensitivity analysis after excluding the events (all-cause death and cardiovascular death) occurring in the first two years of the follow-up. The results showed in similar trends that the SII level in the Q4 group was associated with a 1.33-fold (95% CI, 1.10–1.62) higher risk of cardiovascular mortality and a 1.28-fold (95% CI, 1.17–1.40) higher risk of all-cause mortality compared with the Q1 group. The SIRI level in the Q4 group was also associated with a 1.40-fold (95% CI, 1.14–1.71) higher risk of cardiovascular mortality and a 1.37-fold (95% CI, 1.24–1.51) higher risk of all-cause mortality compared with the Q1 group (Appendix A).

## 4. Discussion

It is gradually becoming more broadly recognized that systemic inflammation initiates and aggravates the pathological process of chronic diseases. A plethora of inflammatory predictors associated with the risk of CVDs and mortality were discovered and targeted treatments were proposed [26]. To the best of our knowledge, this is the first attempt to examine whether the novel SII and SIRI predict cardiovascular and all-cause mortality in the general population in prospective cohorts using the NHANES database, which is a national representative sample. We found that high SII or SIRI were closely linked with increased cardiovascular mortality and all-cause mortality in the general population. The association remained significant after adjusting for confounders. The risk of all-cause death increased evidently in people aged 60 years or older withhigh SII or SIRI levels. Second, the results suggest that the risks endowed by SII and SIRI were changed alongside different race, social status, and traditional risk factors. In short, SII and SIRI can be deemed as effective predictors for assessment of cardiovascular risks in the general population. The screening tool may be used to quickly identify high-risk patients with undesirable health problems and mortality risks, and at a relatively low cost.

A recent meta-analysis supports our conclusion that higher SII was significantly associated with an increased risk of CVDs [19]. This increased risk could be observed in almost all CVDs subtypes. SII levels at the onset of CVDs were significantly higher than those in the general population. Consistently, Hu’s study suggested that SII was associated with increased risk of total stroke, hemorrhagic stroke and ischemic stroke [15]. Likewise, other studies reported that elevation of SII and SIRI increased the risk of hemorrhagic and ischemic stroke subtypes, as well as all-cause death [16]. Furthermore, elevated preoperative SII level is associated with an increased risk of death in individuals with acute ST-elevation myocardial infarction (MI), even though percutaneous coronary intervention was performed in acute MI patients [27,28]. Either SII or SIRI is related to a poor short-term prognosis in atrial fibrillation patients with ischemic stroke. They can be utilized as independent predictors for distinct atrial fibrillation types [20,29,30]. SIRI was recently found to be an independent predictor for functional outcomes in intracerebral hemorrhage [31] and acute ischemic stroke [32]. It is suggested that SII and SIRI may be used to identify individuals at risk for CVDs and facilitate early diagnosis.

Although other studies have found links between cellular types of peripheral leukocytes and cardiovascular events [33,34], we analyze the relationship between each of the novel inflammatory indices and cardiovascular mortality at the same time. Neutrophil, the innate immune cell, is the traditional index to reflect inflammatory status of the immune system. Circulatory monocytes, which can infiltrate solid tissue and become macrophage, participates in immune defense and the damage-repairment process. Conversely, lymphocytes regulate immune system through secretion of cytokines and cytolytic activity. Therefore, SIRI, which is calculated by the counts of peripheral neutrophils, monocytes, and lymphocytes, might be a powerful and reliable indicator to reflect inflammatory condition. On the other hand, SII, the index in consideration of platelets, neutrophils, and lymphocytes, might be a comprehensive indicator to predict the coagulation and inflammation risk of CVDs events. SII and SIRI are thought to be more reliable and representative of inflammation and thrombosis than traditional indicators such as PLR and NLR [35,36]. SII was found to be a better predictor of coronary artery disease than PLR, NLR, and CRP [15]. Unlike the composite indices, single blood cell count can be affected by factors such as alterations in body fluids. In our study, the highest quartile of SII or SIRI participants often showed neutrophilia, thrombocytosis, monocytosis, and lymphocytopenia, indicating a combination of nonspecific inflammation and damage in the adaptive immune response [37]. Changes in microvascular permeability caused by the inflammatory response are known to be essential processes in the pathophysiology of atherosclerosis [38]. Systemic inflammation can cause aberrant platelet aggregation, which can stick to the surface of endothelial cells and induce local ischemia, hypoxia, and microthrombosis, resulting in local tissue death [39,40]. The aberrant fall in lymphocyte count suggests excessive lymphocyte death in the body, which leads to a decline in immune system capabilities and immunological dysfunction. Subsequently, death of lymphocyte facilitates endothelial dysfunction, abnormal aggregation and thrombosis after platelet activation. Monocytes [41] and neutrophils [40] can also promote aberrant coronary plaque state, induce atherosclerotic plaque rupture and thrombosis, and increase the risk of adverse cardiovascular events by activating and producing an inflammatory response. SII and SIRI can efficiently and thoroughly reflect the inflammatory state and immune system status of the body as a composite inflammatory index. The cumulative effect of three different cell lines that influence each other increases the predictive value for CVDs and mortality synergistically. Interactions between platelets, neutrophils, and lymphocytes, as well as neutrophils, monocytes, and lymphocytes, may thus represent novel targets for chronic inflammation underlying CVDs.

In our study, the interaction between SII or SIRI and death risk was not significant in the subgroup analysis by gender. This indicates that after adjusting for confounding factors, the results of the association between higher SII or SIRI levels and mortality remained reliable regardless of gender. Of interest, using age 60 as the watershed, interaction was significant when we analyzed SII or SIRI with all-cause mortality (*p* = 0.008 and 0.003 for interactions, respectively). Age independently influenced the effect of SII or SIRI level on the risk of all-cause mortality. Over 60 years old, persons with higher SII levels or SIRI levels are at higher risk of dying from any cause. Aging confers systemic changes onthe whole body. Mechanistically, age-related increases in inflammatory levels are known as aging-associated inflammatory responses [42]. The elderly, senescent cells, mainly fibroblasts, contribute to the production of proinflammatory factors, collectively called senescence-associated secretory phenotypes [43]. With aging, the immune system is less efficient in degrading misfolded proteins and organelles, and senescent cells accumulate, thus causing systemic inflammation [44].

There are several strengths and limitations to our study. First, it was adequate to provide reliable conclusions and statistical power considering the large-scale sample size and the complex, multistage, probability sampling design included in NHANES; second, we controlled many known risk factors to exclude possible confounders to investigate the value of novel inflammatory indices for mortality prediction; third, owing to large ethnic and cultural differences in diet, physical activity, and genetic susceptibility to the development of CVDs, the conclusions in the present study based on NHANES could be applicable to multiple populations. However, the limitation of this study should be mentioned. Considering specific causes of cardiovascular death, such as stroke and MI, are not classified in detail in the public database, the association between inflammatory indices and specific cardiovascular subtypes cannot be thoroughly investigated.

## 5. Conclusions

CVDs are a group of serious, potentially life-threatening diseases. Despite remarkable advances in prevention and intervention, the rate of death from cardiovascular events has not significantly declined over the past decade. The number of elderly patients with CVDs is still increasing. We analyzed 42,875 adults in the US from NHANES and determined the prediction value of SII or SIRI level on increased all-cause and cardiovascular mortality risks. In our study, high SII or SIRI increased adverse outcomes in the general population according to the cutoff values determined by quartiles. Moreover, the increase in all-cause mortality was more prominent in the elderly aged ≥60 years. Our results shed insight on the value of earlier prevention and innovative treatment of inflammation to reduce CVDs burden. Further studies with larger cohorts are necessary to confirm our findings, to improve the adverse outcomes in the general population. Additionally, the association between these inflammatory indices and specific cardiovascular subtypes needs to be thoroughly investigated.

## Figures and Tables

**Figure 1 jcm-12-01128-f001:**
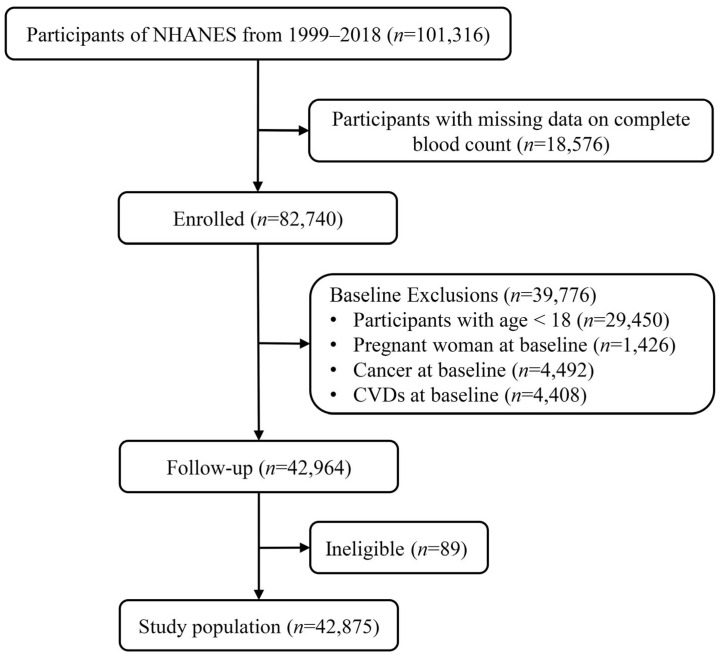
Flow chart of study population selection.

**Figure 2 jcm-12-01128-f002:**
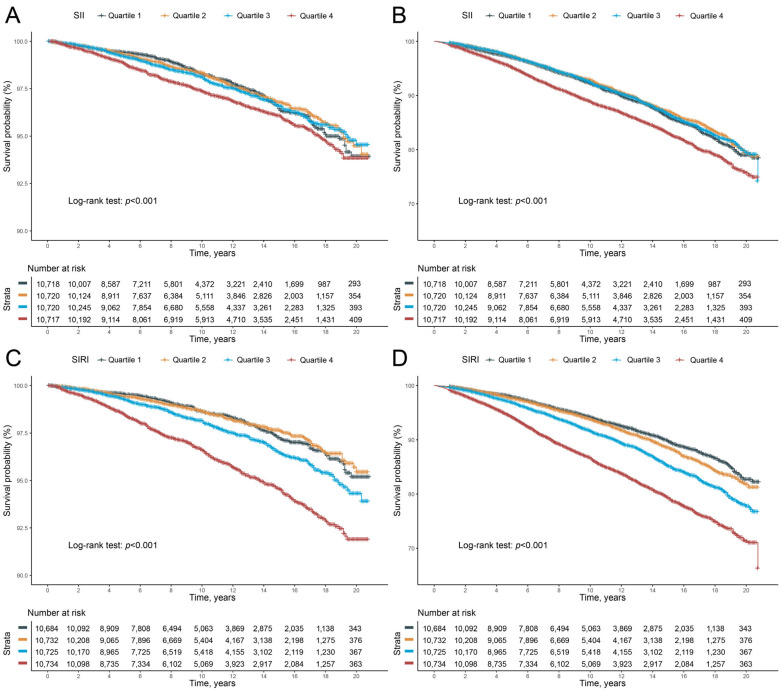
(**A**,**B**) The Kaplan-Meier curves for 20-year occurrence of (**A**) cardiovascular and (**B**) all-cause mortality, by quartiles of SII. (**C**,**D**) The Kaplan-Meier curves for 20-year occurrence of (**C**) cardiovascular and (**D**) all-cause mortality, by quartiles of SIRI.

**Table 1 jcm-12-01128-t001:** Baseline characteristics of quantiles of Systemic immune inflammation index (SII) among the general adult population in the National Health and Nutrition Examination Survey (NHANES), 1999–2018.

Variables	Quartiles of SII				*p* Value
	Q1	Q2	Q3	Q4	
Participants, *n*	10,718	10,720	10,720	10,717	
Age, years	44.47 (18.10)	44.44 (17.80)	44.21 (17.79)	44.69 (18.54)	0.298
Male, %	5895 (55.0)	5453 (50.9)	5108 (47.6)	4627 (43.2)	<0.001
Race/ethnicity, %					<0.001
Mexican American	1736 (16.2)	2210 (20.6)	2323 (21.7)	2309 (21.5)	
Other Hispanic	862 (8.0)	1000 (9.3)	961 (9.0)	904 (8.4)	
Non-Hispanic White	3125 (29.2)	4186 (39.0)	4655 (43.4)	5063 (47.2)	
Non-Hispanic Black	3755 (35.0)	2170 (20.2)	1776 (16.6)	1586 (14.8)	
Other race	1240 (11.6)	1154 (10.8)	1005 (9.4)	855 (8.0)	
Education level, %					0.001
Below high school	2906 (27.1)	2929 (27.3)	2931 (27.3)	2949 (27.5)	
High school	2336 (21.8)	2305 (21.5)	2449 (22.8)	2543 (23.7)	
Above high school	5476 (51.1)	5486 (51.2)	5340 (49.8)	5225 (48.8)	
Poverty, %	2358 (22.0)	2223 (20.7)	2274 (21.2)	2388 (22.3)	0.022
Smoking status, %					<0.001
Never smokers	6451 (60.2)	6386 (59.6)	6240 (58.2)	5831 (54.4)	
Former smokers	2127 (19.8)	2185 (20.4)	2145 (20.0)	2239 (20.9)	
Current smokers	2140 (20.0)	2149 (20.0)	2335 (21.8)	2647 (24.7)	
Drinking status, %					<0.001
Non-drinkers	2430 (22.7)	2436 (22.7)	2242 (20.9)	2446 (22.8)	
Low-to-moderate drinkers	7488 (69.9)	7516 (70.1)	7656 (71.4)	7345 (68.5)	
Heavy drinkers	800 (7.5)	768 (7.2)	822 (7.7)	926 (8.6)	
BMI status, %					0.011
<25.0	3841 (35.8)	3454 (32.2)	3237 (30.2)	3340 (31.2)	
25.0–29.9	3603 (33.6)	3677 (34.3)	3566 (33.3)	3241 (30.2)	
>29.9	3274 (30.5)	3589 (33.5)	3917 (36.5)	4136 (38.6)	
Physical activity, %					<0.001
Inactive	2475 (23.1)	2482 (23.2)	2632 (24.6)	2922 (27.3)	
Insufficiently active	3786 (35.3)	3988 (37.2)	4150 (38.7)	4133 (38.6)	
Active	4457 (41.6)	4250 (39.6)	3938 (36.7)	3662 (34.2)	
Energy intake, kcal/day	1984.00 [1473.37, 2600.50]	1977.00 [1494.00, 2595.00]	1960.50 [1472.00, 2568.62]	1922.50 [1462.50, 2535.50]	<0.001
eGFR, mL/min/1.73 m^2^	100.52 (22.87)	100.14 (22.58)	100.33 (23.06)	99.22 (24.54)	<0.001
ALT, U/L	21.00 [16.00, 29.00]	21.00 [16.00, 29.00]	21.00 [16.00, 29.00]	20.00 [15.00, 27.00]	<0.001
AST, U/L	23.00 [20.00, 28.00]	23.00 [19.00, 27.00]	22.00 [19.00, 27.00]	22.00 [18.00, 26.00]	<0.001
Hypertension, %	2837 (26.5)	2788 (26.0)	2859 (26.7)	3110 (29.0)	<0.001
Diabetes, %	905 (8.4)	966 (9.0)	875 (8.2)	1030 (9.6)	0.001
WBC count, 10^3^/μL	5.80 [4.90, 6.90]	6.60 [5.60, 7.70]	7.20 [6.10, 8.40]	8.20 [6.90, 9.90]	<0.001
Neutrophils count, 10^3^/μL	2.80 [2.20, 3.40]	3.60 [3.00, 4.30]	4.30 [3.60, 5.10]	5.50 [4.60, 6.70]	<0.001
Monocyte count, 10^3^/μL	0.50 [0.40, 0.60]	0.50 [0.40, 0.60]	0.50 [0.40, 0.70]	0.60 [0.50, 0.70]	<0.001
Lymphocyte count, 10^3^/μL	2.30 [1.90, 2.80]	2.20 [1.80, 2.60]	2.00 [1.70, 2.50]	1.80 [1.50, 2.30]	<0.001
Platelet count, 10^3^/μL	211.00 [180.00, 243.00]	238.00 [209.00, 273.00]	260.00 [227.00, 297.00]	294.00 [253.00, 342.00]	<0.001
SIRI	0.57 [0.42, 0.77]	0.87 [0.68, 1.10]	1.14 [0.89, 1.45]	1.69 [1.25, 2.31]	<0.001

Normally distributed continuous variables were described as means and SD, and continuous variables without a normal distribution were presented as medians [interquartile ranges]. Categorical variables are presented as numbers (percentages). eGFR, estimated glomerular filtration rate; ALT, alanine aminotransferase; AST, aspartate aminotransferase; WBC, white blood cell; SII, systemic immune-inflammatory index; SIRI, systemic inflammatory response index.

**Table 2 jcm-12-01128-t002:** Baseline characteristics of quantiles of SIRI among the general adult population in NHANES, 1999–2018.

Variables	Quartiles of SIRI				*p* Value
	Q1	Q2	Q3	Q4	
Participants, *n*	10,684	10,732	10,725	10,734	
Age, years	43.05 (17.06)	43.98 (17.48)	44.58 (17.99)	46.19 (19.47)	<0.001
Male, %	4616 (43.2)	5031 (46.9)	5429 (50.6)	6007 (56.0)	<0.001
Race/ethnicity, %					<0.001
Mexican American	1797 (16.8)	2340 (21.8)	2297 (21.4)	2144 (20.0)	
Other Hispanic	828 (7.7)	1013 (9.4)	975 (9.1)	911 (8.5)	
Non-Hispanic White	2642 (24.7)	4105 (38.3)	4834 (45.1)	5448 (50.8)	
Non-Hispanic Black	4075 (38.1)	2154 (20.1)	1645 (15.3)	1413 (13.2)	
Other race	1342 (12.6)	1120 (10.4)	974 (9.1)	818 (7.6)	
Education level, %					<0.001
Below high school	2795 (26.2)	2969 (27.7)	2941 (27.4)	3010 (28.0)	
High school	2229 (20.9)	2352 (21.9)	2400 (22.4)	2652 (24.7)	
Above high school	5660 (53.0)	5411 (50.4)	5384 (50.2)	5072 (47.3)	
Poverty, %	2348 (22.0)	2270 (21.2)	2258 (21.1)	2367 (22.1)	0.150
Smoking status, %					<0.001
Never smokers	6897 (64.6)	6473 (60.3)	6151 (57.4)	5387 (50.2)	
Former smokers	1905 (17.8)	2203 (20.5)	2177 (20.3)	2411 (22.5)	
Current smokers	1882 (17.6)	2056 (19.2)	2397 (22.3)	2936 (27.4)	
Drinking status, %					<0.001
Non-drinkers	2716 (25.4)	2381 (22.2)	2281 (21.3)	2176 (20.3)	
Low-to-moderate drinkers	7273 (68.1)	7598 (70.8)	7611 (71.0)	7523 (70.1)	
Heavy drinkers	695 (6.5)	753 (7.0)	833 (7.8)	1035 (9.6)	
BMI status, %					<0.001
<25.0	3898 (36.5)	3469 (32.3)	3220 (30.0)	3285 (30.6)	
25.0–29.9	3501 (32.8)	3616 (33.7)	3541 (33.0)	3429 (31.9)	
>29.9	3285 (30.7)	3647 (34.0)	3964 (37.0)	4020 (37.5)	
Physical activity, %					<0.001
Inactive	2520 (23.6)	2560 (23.9)	2549 (23.8)	2882 (26.8)	
Insufficiently active	3927 (36.8)	4078 (38.0)	4084 (38.1)	3968 (37.0)	
Active	4237 (39.7)	4094 (38.1)	4092 (38.2)	3884 (36.2)	
Energy intake, kcal/day	1913.00 [1439.00, 2503.00]	1955.05 [1467.00, 2570.38]	1977.00 [1496.68, 2586.00]	1999.00 [1498.00, 2631.97]	<0.001
eGFR, ml/min/1.73 m^2^	103.09 (22.07)	100.83 (22.09)	99.40 (23.06)	96.91 (25.30)	<0.001
ALT, U/L	20.00 [15.00, 27.00]	21.00 [16.00, 29.00]	21.00 [16.00, 29.00]	21.00 [16.00, 29.00]	<0.001
AST, U/L	22.00 [19.00, 27.00]	22.00 [19.00, 27.00]	22.00 [19.00, 27.00]	22.00 [19.00, 27.00]	0.902
Hypertension, %	2642 (24.7)	2718 (25.3)	2860 (26.7)	3374 (31.4)	<0.001
Diabetes, %	796 (7.5)	908 (8.5)	941 (8.8)	1131 (10.5)	<0.001
WBC count, 10^3^/μL	5.50 [4.70, 6.50]	6.50 [5.60, 7.60]	7.30 [6.30, 8.50]	8.60 [7.30, 10.10]	<0.001
Neutrophils count, 10^3^/μL	2.70 [2.20, 3.20]	3.60 [3.10, 4.20]	4.40 [3.70, 5.10]	5.60 [4.70, 6.70]	<0.001
Monocyte count, 10^3^/μL	0.40 [0.30, 0.50]	0.50 [0.40, 0.60]	0.60 [0.50, 0.70]	0.70 [0.60, 0.80]	<0.001
Lymphocyte count, 10^3^/μL	2.20 [1.80, 2.70]	2.10 [1.70, 2.60]	2.10 [1.70, 2.60]	1.90 [1.50, 2.40]	<0.001
Platelet count, 10^3^/μL	239.00 [203.00, 281.00]	246.00 [210.00, 289.00]	251.00 [214.00, 296.00]	258.50 [218.00, 305.00]	<0.001
SII	291.19 [219.20, 375.67]	413.77 [332.81, 518.14]	523.89 [422.93, 653.14]	748.52 [590.89, 983.48]	<0.001

Normally distributed continuous variables were described as means and SD, and continuous variables without a normal distribution were presented as medians [interquartile ranges]. Categorical variables were presented as numbers (percentages). eGFR, estimated glomerular filtration rate; ALT, alanine aminotransferase; AST, aspartate aminotransferase; WBC, white blood cell; SII, systemic immune-inflammatory index; SIRI, systemic inflammatory response index.

**Table 3 jcm-12-01128-t003:** HRs (95% Cis) of cardiovascular diseases and all-cause mortalities according to quartiles of SII among the general adult population in NHANES, 1999–2018.

	Q1	Q2	Q3	Q4	*p* Trend
Levels of SII	<335.36	355.36–468.83	468.84–655.55	>655.56	
Cardiovascular mortality					
No. deaths/total	203/10,718	222/10,720	249/10,720	324/10,717	
Crude	Reference	1.00 (0.83–1.21)	1.05 (0.88–1.27)	1.32 (1.11–1.57)	0.002
Model 1	Reference	1.12 (0.92–1.35)	1.20 (0.99–1.45)	1.43 (1.19–1.71)	0.001
Model 2	Reference	1.09 (0.90–1.32)	1.16 (0.96–1.40)	1.33 (1.11–1.59)	0.015
All-cause mortality					
No. deaths/total	897/10,718	923/10,720	1008/10,720	1422/10,717	
Crude	Reference	0.94 (0.86–1.03)	0.97 (0.88–1.06)	1.31 (1.21–1.43)	<0.001
Model 1	Reference	1.01 (0.92–1.11)	1.05 (0.96–1.15)	1.38 (1.26–1.50)	<0.001
Model 2	Reference	0.99 (0.91–1.09)	1.03 (0.94–1.13)	1.29 (1.18–1.41)	<0.001

Data are presented as HR (95% CI) unless indicated otherwise.

**Table 4 jcm-12-01128-t004:** HRs (95% Cis) of cardiovascular diseases and all-cause mortality according to quartiles of SIRI among the general adult population in NHANES, 1999–2018.

	Q1	Q2	Q3	Q4	*p* Trend
Levels of SIRI	<0.68	0.68–0.98	0.99–1.42	>1.43	
Cardiovascular mortality					
No. deaths/total	177/10,684	181/10,732	246/10,725	394/10,734	
Crude	Reference	0.98 (0.80–1.21)	1.35 (1.12–1.64)	2.26 (1.89–2.70)	<0.001
Model 1	Reference	0.90 (0.73–1.12)	1.18 (0.96–1.44)	1.60 (1.32–1.94)	<0.001
Model 2	Reference	0.88 (0.71–1.09)	1.10 (0.90–1.34)	1.39 (1.14–1.68)	<0.001
All-cause mortality					
No. deaths/total	737/10,684	851/10,732	1080/10,725	1582/10,734	
Crude	Reference	1.11 (1.01–1.22)	1.43 (1.30–1.57)	2.18 (2.00–2.38)	<0.001
Model 1	Reference	1.01 (0.91–1.12)	1.22 (1.11–1.35)	1.57 (1.43–1.72)	<0.001
Model 2	Reference	0.99 (0.90–1.10)	1.16 (1.05–1.28)	1.39 (1.26–1.52)	<0.001

Data are presented as HR (95% CI) unless indicated otherwise.

## Data Availability

This study analyzed publicly available datasets. These data can be found here: (https://wwwn.cdc.gov/nchs/nhanes/analyticguidelines.aspx, accessed on 1 November 2022).

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
