# Peer review of "Systemic Immune Inflammation Index (SII), System Inflammation Response Index (SIRI) and Risk of All-Cause Mortality and Cardiovascular Mortality: A 20-Year Follow-Up Cohort Study of 42,875 US Adults"

_jcm, 2023, doi:10.3390/jcm12031128_

Round 1

Reviewer 1 Report

Xia et al in this manuscript analyzed the NHANES database and studied the impact of SII or SIRI level on cardiovascular disease risk. It is overall a well written manuscript with conclusion appropriately  based on the results. 

I have a few suggestions:

1.     Introduction is completely lacking references – multiple sentences describing previous literature and evidence needs to be referenced appropriately. 

2.     Infact despite references, there is no references in the manuscript text making it hard to review. 

3.     Comma and not full stop in between the sentences – “Even though great advances have been achieved in the 38 CVDs prevention and treatment. they remain the leading causes of death and disabilities 39 worldwide .”

4.     Definition of SII and SIRI and classification into groups – why was these numbers chosen? Was it based on previous any previous studies/references?

5.     Baseline characteristics of population stratified by SII/SIRI should not limited to supplemental tables as this is the essence of the study. This essentially should be table 1 rather than describing entire population. 

6.     PMID: 36003917 is the most recent metanalysis combining evidence on this topic and needs to be referenced.

7.     This line is not true as noted in above reference – “To the best of our knowledge, this study is the first attempt to investigate whether the novel SII, SIRI indices can predict adverse outcomes in a national representative prspective cohort” This needs to be appropriately corrected. 

Author Response

A point-by-point response to Reviewer 1

We sincerely thank Reviewer 1 for your very constructive comments. We have made amendments on our manuscript as you suggested. Our point-by-point responses are listed below:

Reviewer 1:

Xia et al in this manuscript analyzed the NHANES database and studied the impact of SII or SIRI level on cardiovascular disease risk. It is overall a well written manuscript with conclusion appropriately based on the results.

We sincerely thank you for your positive comments. We have revised our manuscript as you suggested.

Point 1: Introduction is completely lacking references – multiple sentences describing previous literature and evidence needs to be referenced appropriately.

Response 1: Thank you for your very helpful comments and kindly reminding! We are so sorry for the mistake in the introduction, we have cited previous literature and evidence as you suggested.

Point 2: In fact, despite references, there is no references in the manuscript text making it hard to review.

Response 2: Sorry for this technological mistake! Thanks for your useful suggestion! We have revised this issue throughout the manuscript.

Point 3: Comma and not full stop in between the sentences – “Even though great advances have been achieved in the 38 CVDs prevention and treatment. they remain the leading causes of death and disabilities 39 worldwide.”

Response 3: Thank you for your kindly reminding! We have corrected our manuscript.

Point 4: Definition of SII and SIRI and classification into groups – why was these numbers chosen? Was it based on previous any previous studies/references?

Response 4: This is an insightful comment! Thanks for highlighting this significant point! In the present study, we determined the cutoff values for evaluating SII and SIRI levels based on quartiles. In fact, there is no unified standard for grouping SII and SIRI, according to previous reports. In an updated meta-analysis, a total of 13 studies were eligible for analysis. In these studies, the determination of SII cutoff values varied and was based on ROC analysis, Youden index, and quartiles (DOI: 10.3389/fcvm.2022.933913). Recently, the two indicators are often grouped by quartiles in studies of the association of SII and SIRI with cardiovascular events. For example, in Xu’s study, they determined the best cutoff values for evaluating SII levels and the incidence of CVDs in middle-aged and elderly Chinese based on quartiles (DOI: 10.1016/j.atherosclerosis.2021.02.012). In the study of Lin et al., they divided SII and SIRI into four groups based on quartiles, and evaluated the association between the two indices and the adverse prognosis of atrial fibrillation-associated ischemic stroke patients (DOI: 10.1186/s40001-022-00733-9).

Point 5: Baseline characteristics of population stratified by SII/SIRI should not limited to supplemental tables as this is the essence of the study. This essentially should be table 1 rather than describing entire population.

Response 5: Thank you for your useful suggestion! This article studied the impact of SII or SIRI level on cardiovascular disease risk and all-cause mortality. We have adjusted baseline characteristics of population stratified by SII/SIRI to Table 1 and 2. Baseline characteristics of the general population were adjusted to Table S1. We have corrected our manuscript as you suggested.

Point 6: PMID: 36003917 is the most recent metanalysis combining evidence on this topic and needs to be referenced.

Response 6: Thank you for this helpful comment! We have cited this article as you suggested.

Point 7: This line is not true as noted in above reference – “To the best of our knowledge, this study is the first attempt to investigate whether the novel SII, SIRI indices can predict adverse outcomes in a national representative prospective cohort” This needs to be appropriately corrected.

Response 7: Thank you for your very helpful comment and kindly reminding! We have modified the statement according to your suggestion.

Reviewer 2 Report

·   Overall, the paper was well-written and clear.

· But there are several grammatical errors, and some confusing sentences need to rectify. Suggest the manuscript should have been read and edited by a native English speaker with knowledge of the techniques.

· Do not start a sentence with numerals.

· Figure 2: The thickness of the line can be reduce.

· Conclusion too simple - need to revise and highlight the main outcomes of the study.

Author Response

A point-by-point response to Reviewer 2

We sincerely thank Reviewer 2 for your very constructive comments. We have made changes according to your suggestions. Our point-by-point responses are listed below:

Reviewer 2:

Overall, the paper was well-written and clear. But there are several grammatical errors, and some confusing sentences need to rectify.

We sincerely thank you for your positive comments. We have revised our manuscript as you suggested.

Point 1: Suggest the manuscript should have been read and edited by a native English speaker with knowledge of the techniques.

 Response 1: Thank you for your kind comment! We have asked Professor Shao-Liang Chen, who is a well-known expert, to polish our paper. We have revised our manuscript as you suggested.

Point 2: Do not start a sentence with numerals.

Response 2: Thanks! We followed your suggestion. We have revised the relevant sentences.

Point 3: Figure 2: The thickness of the line can be reduced.

Response 3: Thanks for highlighting this significant point! We plotted Kaplan-Meier curves for survival using R software. The lines in different colors represent the four groups of participants according to SII/SIRI distribution. Each vertical bar represents a censored data. We have managed to adjust each line to the finest extent, but the height of the censored data could not be changed. Anyway, it had no influence on the statistical efficacy. We can find that participants in the Quartile 4 group had a worsened prognosis with the longer follow-up time.

Point 4: Conclusion too simple - need to revise and highlight the main outcomes of the study.

 Response 4: Thank you for your helpful suggestion! The main conclusion of our study is that we found that high SII or SIRI were closely linked with increased cardiovascular mortality and all-cause mortality in the general population. The association remained significant after adjusting for confounders. The risk of all-cause death increased obviously in people aged 60 years or older in high SII or SIRI level. We have modified the statement according to your suggestion.

Reviewer 3 Report

In this study entitled “Systemic Immune Inflammation Index (SII), System Inflammation Response Index (SIRI) and Risk of All-Cause Mortality and Cardiovascular Mortality. A 20 Year Follow Up Cohort Study of 42,875 US Adults)” the authors investigated the role of two novel inflammation indices in predicting adverse cardiovascular outcomes. They investigated a large number of patients with a long follow up, obtaining information from the National Health and Nutrition Examination Survey (NHANES).

The authors reported that patients with SII and SIRI at the highest quartile had worse outcomes (increased all cause and cardiovascular mortality) than those in the lowest quartile.

The topic is interesting and support growing evidence that inflammation is an important risk factor and predictor of cardiovascular events.

The authors provided data of good quality. The methodology is well described, and the conclusions are supported by the results.

Here are specific comments:

1)    In the entire article the references are not mentioned in the body text; they are just listed at the end of the paper. This does not allow to understand if a specific sentence has been extracted by previous works or is just a conclusion made by the authors. A new version of the paper with each reference located at the right place in the text is needed.

2)    The authors should provide a structured abstract (“background and aims, methods, results, conclusion).

3)    The authors should better specify in the abstract and in the main text the inclusion and exclusion criteria for patients to be included in the registry.

4)    A discussion/comparison about other evidence regarding other inflammatory markers and predictors would be appreciated. In particular what could be the advantages and disadvantages of the two new proposed indices compared to the previous ones.

5)    The English stile should be revised: there are several typos, misspellings, and other errors.

Author Response

A point-by-point response to Reviewer 3

We sincerely thank Reviewer 3 for your very constructive comments. We have made changes according to your suggestions. Our point-by-point responses are listed below:

Reviewer 3:

In this study entitled “Systemic Immune Inflammation Index (SII), System Inflammation Response Index (SIRI) and Risk of All-Cause Mortality and Cardiovascular Mortality. A 20 Year Follow Up Cohort Study of 42,875 US Adults)” the authors investigated the role of two novel inflammation indices in predicting adverse cardiovascular outcomes. They investigated a large number of patients with a long follow up, obtaining information from the National Health and Nutrition Examination Survey (NHANES).

The authors reported that patients with SII and SIRI at the highest quartile had worse outcomes (increased all cause and cardiovascular mortality) than those in the lowest quartile.

The topic is interesting and support growing evidence that inflammation is an important risk factor and predictor of cardiovascular events.

The authors provided data of good quality. The methodology is well described, and the conclusions are supported by the results.

 We sincerely thank you for your positive comments. We have revised our manuscript as you suggested.

 Point 1: In the entire article the references are not mentioned in the body text; they are just listed at the end of the paper. This does not allow to understand if a specific sentence has been extracted by previous works or is just a conclusion made by the authors. A new version of the paper with each reference located at the right place in the text is needed.

 Response 1: Sorry for this mistake! Thanks for your kindly reminding! We have revised this issue throughout the manuscript.

 Point 2: The authors should provide a structured abstract (“background and aims, methods, results, conclusion).

 Response 2: Thanks for your kind suggestion! We have revised our manuscript as you suggested.

Point 3: The authors should better specify in the abstract and in the main text the inclusion and exclusion criteria for patients to be included in the registry.

 Response 3: Thanks! We followed your suggestion. We have described the inclusion and exclusion criteria for participants more clearly in more in the abstract and in the main text.

Point 4: A discussion/comparison about other evidence regarding other inflammatory markers and predictors would be appreciated. In particular what could be the advantages and disadvantages of the two new proposed indices compared to the previous ones.

 Response 4: This is a very constructive suggestion! According to previous reports, SII and SIRI are thought to be more reliable and representative markers of inflammation and thrombosis than traditional indicators such as PLR and NLR (DOI: 10.1111/eci.13230; DOI: 10.3389/fneur.2022.836595). Unlike single blood cell count, SII or SIRI, which combines three different subsets of blood cells to reflect inflammation and thrombosis, is not significantly affected by factors such as alterations in body fluids. Furthermore, SII/SIRI has the advantage of easy availability and rapidity, because routine blood analysis is essential for patients admitted to the hospital, and patients are not required to pay extra costs, thus improving compliance. Due to the nature of observational studies, neither intervention strategies are compared nor treatment suggestions are proposed in our study. In addition, future studies should further determine the optimal cutoff value, characterize the benefited population, and investigate whether SII/SIRI can be used in conjunction with other known CVDs risk factors to build an accurate CVDs risk assessment system.

 Point 5: The English stile should be revised: there are several typos, misspellings, and other errors.

 Response 5: Thank you for your kind comment! We have found and corrected the typos, misspellings, and other errors. We have revised our manuscript as you suggested.